# Development and In Vitro Differentiation of Schwann Cells

**DOI:** 10.3390/cells11233753

**Published:** 2022-11-24

**Authors:** Sarah Janice Hörner, Nathalie Couturier, Daniele Caroline Gueiber, Mathias Hafner, Rüdiger Rudolf

**Affiliations:** 1Institute of Molecular and Cell Biology, Mannheim University of Applied Sciences, 68163 Mannheim, Germany; 2Interdisciplinary Center for Neurosciences, Heidelberg University, 69120 Heidelberg, Germany; 3Center for Mass Spectrometry and Optical Spectroscopy, Mannheim University of Applied Sciences, 68163 Mannheim, Germany; 4Department of Electronics Engineering, Federal University of Technology Paraná, Ponta Grossa 84017-220, Brazil; 5Institute of Medical Technology, Heidelberg University and Mannheim University of Applied Sciences, 69117 Heidelberg, Germany

**Keywords:** differentiation, glia, hiPSC, MSC, neural crest, Schwann cells, stem cells, tissue engineering

## Abstract

Schwann cells are glial cells of the peripheral nervous system. They exist in several subtypes and perform a variety of functions in nerves. Their derivation and culture in vitro are interesting for applications ranging from disease modeling to tissue engineering. Since primary human Schwann cells are challenging to obtain in large quantities, in vitro differentiation from other cell types presents an alternative. Here, we first review the current knowledge on the developmental signaling mechanisms that determine neural crest and Schwann cell differentiation in vivo. Next, an overview of studies on the in vitro differentiation of Schwann cells from multipotent stem cell sources is provided. The molecules frequently used in those protocols and their involvement in the relevant signaling pathways are put into context and discussed. Focusing on hiPSC- and hESC-based studies, different protocols are described and compared, regarding cell sources, differentiation methods, characterization of cells, and protocol efficiency. A brief insight into developments regarding the culture and differentiation of Schwann cells in 3D is given. In summary, this contribution provides an overview of the current resources and methods for the differentiation of Schwann cells, it supports the comparison and refinement of protocols and aids the choice of suitable methods for specific applications.

## 1. Introduction

Schwann cells are the most abundant cell type in the peripheral nerves and the best studied type of peripheral glia cells [1,2,3]. They are most well known for their function of wrapping myelin sheaths around peripheral axons, but the continuously increasing knowledge about Schwann cell biology has revealed a plethora of additional roles besides myelination and neurotrophic support [4], such as regulation of sensory perception [5,6], synaptic communication [7,8], and immune response [9,10]. They are also important in nerve repair, thus contributing to the regenerative ability of the peripheral nervous system (PNS) [11,12]. Therefore, models for exploring the physiology and pathophysiology of Schwann cells in vitro are of profound interest [13]. At the same time, Schwann cells are highly interesting for tissue engineering, since they play a critical role in nerve regeneration upon injury and are affected in peripheral neuropathies such as Charcot-Marie-Tooth disease and Guillain-Barré syndrome [11,14,15,16]. Therefore, the potential of Schwann cells regarding regenerative cell transplantation has been intensely explored in rodent models, with encouraging results. However, for potential clinical applications in human injury and disease, safe and reproducible methods for obtaining large quantities of human Schwann cells are needed. Primary human Schwann cells can be isolated from peripheral nerves, but obtaining a sufficient amount of cells to enable treatment is challenging. In addition, laborious methods are required to establish cultures, and these have a finite capacity for proliferation and passaging [17,18]. As such, rodent Schwann cells have mainly been used for in vitro studies. These are relatively easy to culture, do not cease proliferating in vitro, and can therefore be expanded to high numbers. They also achieve higher degrees of maturation in culture and are able to efficiently myelinate cocultured neurons, while human Schwann cells show only a poor myelination capacity in coculture assays [17,18,19,20]. As these differences hamper work with human Schwann cells and as data from non-human Schwann cells might not be translatable, alternative human in vitro models are important, to obtain valid information regarding Schwann cell development, pathology, and therapeutic potential. Advancements in stem cell technology have made in vitro derivation from multipotent cell sources a promising alternative [21,22,23], and some of these techniques have been used to differentiate Schwann cells in vitro, including human embryonic stem cells (hESC) and human induced pluripotent stem cells (hiPSC), as well as human and non-human adult tissue-derived stem cell types. However, many of these methods still suffer from drawbacks such as incomplete differentiation efficiency, variable reproducibility, or a lack of maturity of the derived cells. Starting from the current knowledge on the embryonal development of the Schwann cell lineage and the underlying molecular determinants, this literature review provides a detailed overview of the differentiation methods and protocols available to derive Schwann cells in vitro from different multipotent cell sources, with a focus on hiPSC/hESC. It compares and discusses the mechanisms, advantages, and drawbacks of the protocols, and summarizes characterization methods and standards. Lastly, it gives a brief insight into methods and trends in the rapidly progressing field of 3D cell culture of Schwann cells.

## 2. In Vivo Development of Schwann Cells

### 2.1. Development of Neural Crest and Schwann Cell Precursors

The Schwann cell lineage arises from the neural crest, a multipotent stem cell population specific to vertebrate embryonal development (Figure 1A). Of ectodermal origin, the neural crest is induced during gastrulation [24,25] and subsequently specified from the neural plate border territory by signaling from adjacent developing structures, i.e., the juxtaposed non-neural ectoderm, neural plate, and underlying mesoderm. After closure of the neural tube, premigratory neural crest cells undergo an epithelial-to-mesenchymal transition (EMT), allowing them to delaminate from the neuroepithelium and to migrate through the embryonal tissue. Dorsoventrally emigrating neural crest cells associate with developing peripheral nerves and are then termed Schwann cell precursors [26,27]. Both neural crest cells and Schwann cell precursors reach their target tissues through specific migration pathways and differentiate towards their terminal fate once settled in the target tissues, where they often undergo a mesenchymal-to-epithelial transition [28]. During this migration phase, Schwann cell precursors move along the peripheral nerves.

Neural crest induction and specification are primarily mediated by four major developmental pathways, i.e., Wnt, FGF, BMP, and Notch signaling (for review, see Prasad et al., 2019 [29]). Wnt signaling is crucial for important aspects of neural crest induction, i.e., formation of the neural plate border and subsequent activation of neural crest-specifying transcription factors (reviewed by Ji et al., 2019 [30]). FGF signaling has been especially implicated in the early phases of neural crest specification [24,31]. BMP signaling appears to be critical after initial neural crest induction [32]. It is required at an intermediate activity level, which is provided by gradients of laterally secreted BMP activating signals and medially secreted BMP antagonizing signals [33,34,35]. 

Finally, Notch signaling is relevant in neural crest induction, migration, differentiation, and inhibition of neurogenesis [36,37,38,39], but its exact role in these processes and the apparent variations between species remain to be elucidated [29]. Although a wealth of in vivo and in vitro studies in different species have established these pathways and their crosstalk as principal regulators of neural crest development, new insights about their precise spatiotemporal activities are still emerging. In addition, other pathways have been suggested to play a role in neural crest development, such as retinoic acid signaling [40,41] and the Hippo/YAP pathway. For example, recent findings suggest that Hippo/YAP promotes neural crest fate and migration, as well as fate specification of neural crest cells (reviewed by Zhao et al., 2021 [42]): YAP activation could benefit the EMT and migration of neural crest cells, and studies have suggested the crosstalk of YAP signaling with that of BMP and Wnt [43], retinoic acid [44], and Notch [45] during neural crest development.

Overall, an astounding variety of cell types of neural, as well as mesenchymal, fates are generated from the neural crest [46]. Besides Schwann cells, these also include other peripheral glia such as enteric glia and ganglionic satellite glia, as well as neurons of the peripheral nervous system (PNS), including sensory neurons, postganglionic sympathetic and parasympathetic neurons, and enteric neurons. Furthermore, melanocytes, endocrine cells, and a variety of mesenchymal cells, including cranial bone and connective tissue, are also neural crest descendants. Some of these fates depend on the neural crest anteroposterior axial identity (cranial, vagal, trunk, sacral), as they contribute to different tissue types, suggesting there is already heterogeneity of differentiation competence in the neural crest population [47,48]. While neuron and glia cells differentiate from the neural crest at all axial levels, the major part of Schwann cells and PNS neurons are derived from trunk neural crest [27,49,50,51,52,53].

The development of neural crest cells into mature Schwann cells proceeds through intermediate stages: first, the previously mentioned Schwann cell precursors, which are found in the early embryonic nerve; and then the immature Schwann cells in late embryonic and perinatal nerves (Schwann cell precursor development has been reviewed in detail elsewhere [1,49,54,55]) (Figure 1B). The extrinsic signals inducing and specifying these stages are still unknown [54,56], but a key factor is neuregulin-1 (Nrg1), specifically the axon-derived splicing isoform Nrg1 type III [57,58,59]. In the Schwann cell precursor stage, membrane-bound Nrg1 type III on juxtaposed sensory and motor axons acts as a crucial survival factor through ErbB family receptor tyrosine kinases, mainly ErbB2/3 heterodimers [60,61]. The transition from neural crest to Schwann cell precursors has been connected to the acquisition of typical glia features. In particular, they enter developing nerves, associate with axons, and depend on axonal signals for survival. At the same time, glial and Schwann cell lineage specific markers are upregulated [1]. Nevertheless, Schwann cell precursors retain a high degree of multipotency, and become primed towards immature Schwann cells or other fates in a gradual manner [55,62]. 

Notably, several neural crest-derived cell types outside the Schwann cell lineage are derived from both the neural crest and Schwann cell precursors. One example is the melanocytes, which differentiate either directly from neural crest cells or from Schwann cell precursors, following their migration along nerves [63,64]. A similar mechanism might possibly be true for enteric glia, as both neural crest cells and Schwann cell precursors were found to populate the gut to differentiate [65]. To date, further cell types which can develop from Schwann cell precursors include endoneurial fibroblasts [66]; postganglionic parasympathetic neurons [67,68]; enteric neurons [69]; neuroendocrine chromaffin cells of the adrenal medulla [70] and Zuckerkandl organ [71], a small population of sympathetic neurons in paraganglia [71]; dental pulp mesenchymal stromal cells [72]; and even chondrocytes and osteocytes [73]; as well as, possibly, recently discovered immune-regulating splenic glia [74,75,76]. This remarkable developmental potential of multipotent Schwann cell precursors clearly indicates that no terminal commitment to the Schwann cell lineage has happened at this stage. Indeed, recent work using single-cell transcriptomics has confirmed that early Schwann cell precursors are highly similar to late migratory neural crest cells and exhibit multipotency towards various cell fates, while late Schwann cell precursors that progress towards the Schwann cell fate become gradually more restricted [62]. 

### 2.2. Commitment to Schwann Cell Lineage and Differentiation of Subtypes

Commitment to immature Schwann cells follows the multipotent Schwann cell precursor stage (Figure 1B). After migratory Schwann cell precursors have reached their target tissue, they either detach from nerves and assume other cell fates or further develop into post-migratory immature Schwann cells in a continued association with axons [55,77,78]. This is accompanied by upregulation of glial and Schwann cell markers (e.g., Krox20, P0, S100b, Oct6) and downregulation of earlier markers (e.g., Ap2α, Pax3, Sox2) [3,26,79,80,81]. At the same time, these cells start expressing basal lamina proteins, as well as growth factors, which act on differentiating neurons but also as autocrine survival factors [1,52,82,83]. During this period, immature Schwann cells initiate radial sorting of axons, whereby groups of immature Schwann cells bundle several axons together, before segregating the larger axons designated for myelination; these processes are orchestrated by axonal Nrg1 [84]. Furthermore, Notch signaling seems to be involved in the transition to immature Schwann cells, through boosting Nrg1 signaling; indeed, activation of Notch1 receptor through Jagged1 expressed on adjacent neurons maintains high expression levels of ErbB2 receptors in differentiating Schwann cells. At later stages, Notch inhibits myelination and is suppressed through Krox20 at the onset of the myelination program [85]. While immature Schwann cells are initially highly proliferative, more than Schwann cell precursors [86], they gradually exit the cell cycle during maturation, when Schwann cells assume mature myelinating and non-myelinating phenotypes. This mainly occurs postnatally in rodents [3,86,87]. 

The two known major subtypes of mature Schwann cells are myelinating Schwann cells and non-myelinating Remak Schwann cells. While the first associate with one individual axon through radial sorting and build up the myelin sheath wrapping a segment of this axon, the latter associate with several small diameter axons, and ensheath them in so-called Remak bundles, without producing myelin. Terminal (or perisynaptic) Schwann cells are a third distinct subtype of Schwann cells. They are associated with neuromuscular junctions, and although they are non-myelinating, terminal Schwann cells differ considerably from Remak Schwann cells in terms of their marker expression, appearance, and function [7,88,89]. Besides these, it is likely that other glial subtypes that have not been well characterized so far also arise from Schwann cell precursors and might differentiate through the immature Schwann cell stage as well [79], such as the glia cells of cutaneous sensory corpuscles [90,91]. While some factors that promote the development of the mature myelinating Schwann cell subtype are known, the differential mechanisms that specify the mature non-myelinating fates, including Remak and terminal Schwann cells, are largely unclear.

Myelinating Schwann cells develop through a further intermediate state, termed a promyelinating Schwann cell, which establishes a 1:1 relationship with an axon sorted for myelination [84]. Axonal sorting and maturation of myelinating Schwann cells is dependent on axonal contact and Nrg1 expression levels [92]. Myelination and terminal differentiation appear to involve activation of the adhesion G-protein coupled receptor GPR126 by basal lamina components such as laminin-211 [93,94,95,96]. GPR126 might signal through multiple downstream effectors, one being the myelination onset regulator Krox20 [97] through cAMP-dependent PKA activation [98]. Furthermore, the axonal Nrg1 expression level seems to tune the regulation of myelination fate. Indeed, while higher Nrg1 levels are required for myelination, lower Nrg1 levels may induce Remak differentiation [92]. At the same time, the Nrg1 expression level is proportional to the axon diameter, so that large caliber axons are sorted for myelination and small caliber axons are ensheathed by Remak bundles [99]. In addition, GABA-B receptors were found to regulate development towards non-myelinating Schwann cells through negative modulation of adenylate cyclase and downregulation of myelin-related proteins [100,101,102]. Furthermore, GPR126 is also involved in Remak Schwann cell differentiation [95]. However, it is unknown how GPR126 functions mechanistically in this context and how its role differs from that in differentiation of myelinating Schwann cells. As for terminal Schwann cells, to date, no differential specifying molecular factors are known, and even the developmental time window when they diverge from the other subtypes remained unknown until recently. Using single-cell transcriptomics, Kastriti and colleagues analyzed developmental transitions and fate choices along the neural crest and Schwann cell lineages. This revealed that immature Schwann cells develop into four terminally differentiated cell types: terminal, non-myelinating (Remak), and myelinating Schwann cells, as well as endoneurial fibroblasts. Of these, the terminal Schwann cell fate is the first to diverge from the others; thus, even before the path of endoneurial fibroblasts splits from myelinating and Remak Schwann cells (Figure 1B) [62].

Even after terminal differentiation into mitotically quiescent mature Schwann cell subtypes, the Schwann cell linage retains a remarkably plastic reprogramming potential, and terminal differentiation is reversible. Predominantly as a reaction to nerve injury, mature Schwann cells of all subtypes can re-enter the cell cycle, de-differentiate, and assume what is called the repair phenotype. This is distinct from other mature, as well as developmental, Schwann cell phenotypes [12,103]. Repair Schwann cells are crucial for nerve repair and account for the outstanding regeneration ability of the PNS [11,104]. Following the resolution of nerve damage, repair Schwann cells re-differentiate into mature myelinating and non-myelinating Schwann cells. During this re-differentiation process, prior mature myelinating Schwann cells can assume a mature non-myelinating phenotype and vice versa [105,106]. Notably, they might even re-differentiate into non-glial cells, including melanocytes [64], dental pulp mesenchymal stem cells [72], or enteric neurons [107]. This remarkable plasticity of developing and mature descendants of the Schwann cell lineage is controlled by transcription factor networks and chromatin remodeling [108,109,110] and makes Schwann cells exceptionally interesting for in vitro modeling studies on nerve injury, disease, and regeneration. However, their extensive developmental multipotency and adult plasticity also add to the challenges in deriving, characterizing, and maintaining Schwann cells in vitro. 

## 3. In Vitro Differentiation of Schwann Cells

### 3.1. Molecular Mechanisms

#### 3.1.1. Neural Crest Induction

As expected, in vitro differentiation of Schwann cells capitalizes on the knowledge of the developmental signaling events required to induce neural crest and, subsequently, Schwann cell fate, as described in the previous sections. Therefore, pluripotent stem cells have been manipulated by a corresponding sequence of molecular factors to induce and steer their differentiation (Figure 2). To date, most available protocols can be divided into an initial neuralization and neural crest induction phase, followed by a second phase dedicated to specifying and maturing Schwann cells. In this section, we will discuss the in vitro neural crest induction mechanisms that have been used, in combination with subsequent Schwann cell differentiation. 

Generally, current protocols can be grouped into two different approaches to induce neural crest. Either stem cells are cultured as embryoid bodies in suspension in the presence of neuralizing factors, to induce neurospheres or, alternatively, neural identity is induced in monolayer culture by combining TGFβ/Activin/Nodal signaling inhibition and activation of Wnt signaling (via inhibition of GSK-3β to stabilize β-catenin). The earliest protocols for deriving Schwann cells from hiPSC/hESC made use of the neuralization-inducing activity of stromal cells using direct coculture or a conditioned medium [111,112,113,114,115]. This was mostly done in combination with neurosphere culture and the simultaneous or subsequent addition of the molecules EGF and bFGF (also known as FGF-2 or FGF-β). More recently, the neurosphere induction method has also been used with defined media instead of stromal cell coculture [116,117]. In general, however, the neurosphere induction technique resulted in mixed neural precursor cultures and, thus, necessitated purification steps before ensuing differentiation. In comparison, the monolayer TGFβ block/Wnt activation approach can yield sufficient quantities of neural crest cells, without the need for purification methods [118,119,120,121]. 

The modulation of Wnt signaling drives neural regional patterning in a dose-dependent manner [122,123] and is required for specification of neural crest (reviewed by Ji et al., 2019 [30]). Stimulation of Wnt activity is intended to drive neural crest fate from a neuroectodermal intermediate, which is, in turn, induced by TGFβ signaling inhibition [124,125,126,127,128,129]. Together with TGFβ inhibitor SB431542, some Schwann cell differentiation studies have used a BMP pathway inhibitor (LDN193189) [119,120]. This procedure is known as dual SMAD inhibition, a combination widely applied to induce neuroectoderm [128,130]. In contrast, other studies did not employ a compound to interfere with BMP signaling [121,131]. While the latter reported high neural crest cell yields, without the need for cell purification, FACS was needed to purify the Schwann cell precursors in the presence of BMP inhibition [119] or the cell yield after differentiation was not reported [120]. Thus, there seems to be a contradiction between certain in vitro differentiation results and current knowledge of neural crest development in vivo, which clearly requires BMP signaling.

Indeed, in the developing embryo, mediolateral gradients of both BMP and Wnt signaling are created through a balance between activators secreted from the lateral regions and inhibitors secreted from medial regions. In these gradients, neural crest cells originate from the neural plate border territory under intermediate levels of BMP and Wnt signaling activity [28,132,133]. Thus, the ambiguous literature, which either used BMP inhibition [126,128,130,134] or not [114,129,135,136,137], and reported rare occasions on which BMP agonists helped [138,139] or blocked neural crest induction [137], suggests that BMP signaling is required at a distinct level. This might be affected by the culture-inherent BMP activity [125]. This hypothesis was further addressed by Hackland and colleagues, who suggested that variations in endogenous BMP ligand production levels across hPSC lines and culture systems account for these variable and partly contradictory results [140]. They proposed mediating a controlled activation of BMP signaling by combining a saturating concentration of exogenous BMP4 with simultaneous addition of a BMP antagonist, DMH1, in a defined, sub-maximal concentration, to precisely tune an optimal BMP activity, in a top-down manner [140,141]. With this method, consistent results of robust neural crest induction across different hPSC lines were achieved. Moreover, integration of this approach into a Schwann cell differentiation protocol led to an increased yield and robustness of neural crest induction [118]. In the future, similar signaling-tuning approaches may further aid in neural crest in vitro differentiation and in developing more effective and robust Schwann cell differentiation protocols. This might be also relevant for Wnt signaling. Indeed, while the evidence seems clear that Wnt activation is crucial for deriving neural crest in vitro, early protocols and some studies using the neurosphere induction approach reported neural crest cell derivation without any exogenous Wnt-activating molecule [117,130,134,142]. Notably, these protocols all used dense cell structures, such as embryoid bodies or confluent cultures, where intrinsic autocrine and/or paracrine Wnt signaling could play a role, as demonstrated by Leung and colleagues in high-density cultures [125]. Moreover, all these protocols reported low yields and a need for purification of neural crest cells. In vivo, Wnt signaling is activated to an intermediate level for neural crest induction, comparable to BMP signaling. Thus, some studies have titrated Wnt activators to find the optimum concentration for their system [124,137,143]. However, endogenous fluctuations of Wnt do not seem to be as heterogenous as is the case for BMP activity, since most of the available protocols seem to work with similar concentrations of Wnt agonist across cell lines [119,120,121,125,126,135]. Nevertheless, further refinement testing of the ideal concentration of Wnt agonist for each chemical batch and cell line could be beneficial, as differing optimal concentrations of the widely used Wnt inducer CHIR99021 between batches [124] and hPSC cell lines [143] were reported. Furthermore, Gomez et al., showed that the axial identity of in vitro hPSC-derived neural crest could be modulated by the magnitude of the Wnt signal [144], and that a concentration of 3 µM CHIR99021 induced neural crest cells of anterior character, while increasing concentrations had a gradually posteriorizing effect. Intermediate concentrations of 4 µM to 6 µM were found to suppress neural crest induction, and posterior neural crest differentiation was seen between 7 µM and 12 µM. Notably, intermediate concentrations of Wnt activator, which were not able to induce neural crest cell fate, seemed to induce neuromesodermal-like progenitor cells when combined with FGF activation [143], which could potentially further differentiate into neural crest cells with trunk identity [143,144,145,146]. Others used the lower CHIR99021 concentration of 3 µM and checked for induction of genes associated with defined axial levels. They also reported the anterior character of the obtained neural crest cells [125,126] if not combined with posteriorizing signals [135]. This binary Wnt activity-dependent cranial/trunk identity decision was confirmed by Hackland and colleagues; however, they found that a concentration of 3 µM CHIR99021 already induced trunk identity in their system, and the CHIR99021 concentration had to be reduced to 0.8 µM to induce cranial identity [145]. As a possible reason for this discrepancy, it was suggested that the Hackland protocol was BSA free, while most other protocols, including the study of Gomez and colleagues, used BSA in the medium. Possibly, BSA suppresses the potency of small molecules such as CHIR99021. This further emphasizes that, ideally, concentrations of Wnt agonists should be titrated for each system, to determine the required concentration. Schwann cell differentiation protocols that include Wnt activation usually apply 3 µM CHIR99021, with additional BSA or BSA-containing supplements such as KSR or B27 [119,120,121], and therefore likely generate a neural crest intermediate of anterior identity. However, axial identity, so far, has not been considered. Although, in principle, neural crest cells of all axial levels have the potential to differentiate into Schwann cells, it would be interesting to determine the effect of in vitro neural crest axial identity on the outcome of Schwann cell differentiation. Recent advances in neural crest in vitro derivation have shown an increasingly precise achievement of different axial identities by posteriorizing through modulation of retinoic acid, FGF, and/or BMP signaling (reviewed by Cooper et al., 2022 [147] and Rocha et al., 2020 [50]), beyond the apparent Wnt-mediated basal cranial/trunk decision [144,145]. As knowledge about in vitro neural crest differentiation progresses, this should be integrated into Schwann cell differentiation studies, to advance these methods.

#### 3.1.2. Schwann Cell Specification

During in vivo development, Schwann cells and their precursor stages are specified by signaling from surrounding tissues, and for Schwann cell development after the migratory neural crest phase, the interactions with neuronal axons and extracellular matrix cues are most important (reviewed by Wilson et al., 2020 [148]). While some of these have been identified and targeted with in vitro differentiation protocols, our picture of the signaling mechanisms of Schwann cell specification, diversification, and maturation is likely far from complete. Historically, the first hints about the survival and differentiation factors for Schwann cells came from primary cultures, mostly of rodent origin. Although the signaling components and mechanisms initially remained unclear, in vitro cultures of embryonal neural tube explants demonstrated migrating neural crest cells and the subsequent appearance of early Schwann cell markers, which were dependent on paracrine neural tube tissue-derived factors [149]. Further studies using primary embryonal developing Schwann cells showed that aspects of Schwann cell differentiation were impaired by withdrawal of axonal contact but could be rescued by elevating intracellular cAMP levels [150,151,152]. In addition, cultures of isolated rat embryonal neural crest cells demonstrated glial differentiation potential upon incubation with the adenylate cyclase activator forskolin in medium containing serum [153]. Both serum supplementation and the addition of cAMP elevating agents have been used in most in vitro Schwann cell differentiation studies to date. Around the same time, and also in primary cultures, Nrg1 type III was discovered as a crucial differentiation and survival factor for Schwann cells [20,57,59,61,154,155,156,157]. Indeed, Nrg1/ErbB signaling seems to be a main determinant for Schwann cell fate and to be implicated in all major steps of Schwann cell development, including induction of Schwann cell precursors from neural crest, Schwann cell precursor survival and proliferation, as well as the specification, survival, and maturation of Schwann cells [78,92,99,158,159,160,161]. Therefore, Nrg1 is included in almost all Schwann cell in vitro differentiation studies—usually supplemented as a soluble recombinant peptide that contains the highly conserved Nrg1 EGF-like domain responsible for ErbB activation [19]—and it seems to be crucial for differentiation success and efficiency.

Protocols to differentiate Schwann cells in vitro from hiPSC and/or hESC (further described in Section 3.3.1) have typically employed serum and Nrg1 in the Schwann cell differentiation phase, and varying combinations of forskolin, ascorbic acid, ROCK inhibitors, PDGF, retinoic acid, and bFGF [111,112,120,121,131]. This factor cocktail is partially based on earlier protocols for adult mesenchymal stem cells (MSC) (further described in Section 3.3.2). Of these, many used the same factor cocktail, consisting of a short incubation with β-mercaptoethanol and retinoic acid, followed by a combination of FBS, forskolin, bFGF, PDGF, and Nrg1 [162,163]. Additionally, in some hiPSC/hESC protocols, a combination of CNTF, db-cAMP, and Nrg1 with bFGF [114,117] or without [116] was used, while one protocol applied only db-cAMP, ascorbic acid, and FBS [119]. Unfortunately, the presence of FBS or serum derivatives is still necessary in most of these protocols, at least for some phases of differentiation, and completely serum-free cultures lead to low cell survival [118,121]. Although serum does not seem to be required for cell propagation in primary human Schwann cell cultures, it is regularly used to promote their survival [19]. 

From the aforementioned, it becomes evident that cAMP elevating agents (such as forskolin or db-cAMP) are the most widely used molecules in Schwann cell differentiation protocols, next to Nrg1. Indeed, cAMP signaling has been implicated in Schwann cell development and maturation, as well as myelin formation. Some mechanisms of action for Nrg1 and cAMP on Schwann cell maturation and specification have already been described, in Section 2.2. In addition, cAMP-stimulating agents seem to synergistically increase the effects of factors such as Nrg1, PDGF, or bFGF in vitro, possibly through increasing trophic factor receptor levels [164,165,166]. Furthermore, they induce a differentiated post-mitotic state, as well as myelination onset markers, in primary isolated Schwann cells [164,167,168]. While combined Nrg1 and cAMP elevating signals seem to induce early markers of the myelination program in cultured Schwann cells [169], myelin production in neuronal cocultures requires the presence of ascorbate, which promotes Schwann cell basal lamina assembly and pro-myelin gene transcription [167,170,171]. Hence, ascorbic acid is supplemented for in vitro myelination studies with hiPSC/hESC-derived Schwann cells [111,112,119,121]. 

When MSC are used as a cell source, commonly a short pre-differentiation incubation with a combination of β-mercaptoethanol and retinoic acid is applied, as this step seems to promote differentiation towards neural identities from MSC [163,172,173]. In the case of the neurosphere induction technique, pre-incubation with β-mercaptoethanol is not necessary [174]. Even during the Schwann cell fate induction phase of hiPSC/hESC protocols, a few studies added β-mercaptoethanol followed by retinoic acid, although it can be reasoned that a step inducing a lineage switch is not needed here, as it is for MSC. However, it might well be that these agents play another role here, as β-mercaptoethanol is sometimes used in hPSC differentiation protocols, as a reducing agent, and retinoic acid is a neural morphogen with numerous roles, including—but not limited to—posteriorization. Moreover, Schwann cells already express retinoid receptors during development, and retinoic signaling can have different effects on Schwann cells, such as migration or regulation of myelination; however, it can also inhibit Schwann cell growth [175,176]. 

Next, bFGF acts as mitogen and survival factor for Schwann cell precursors [26,155] and seems to promote transition to immature Schwann cells and their proliferation [78,177,178]. PDGF, specifically the isoform PDGF-BB, promotes survival of Schwann cell precursors, as well as of Schwann cells [26,179,180], and acts as Schwann cell mitogen [178]. Therefore, these factors might mainly act through promoting the proliferation and survival of differentiating cells of the Schwann cell lineage. With the transition from Schwann cell precursors, immature Schwann cells develop autocrine survival circuits [1], and PDGF-BB belongs to those known survival factors, along with others such as IGF or NT3 [83]. In addition, more mature Schwann cells can also express CNTF, which is also used in some differentiation protocols. However, rather than in Schwann cell differentiation, this cytokine has primarily been linked to motoneuron survival [181], as well as the survival, proliferation, and myelination of oligodendrocytes [182,183,184]. Nevertheless, CNTF was shown to induce Krox20 in Schwann cells in DRG explant cultures when combined with bFGF [185]. Interestingly, primary cultures of Schwann cells show density-dependent survival, due to these autocrine signaling circuits [56,78,83,164], and better survival in higher culture densities was mentioned for hiPSC-derived [118], as well as MSC-derived Schwann cells [186]. This suggests that autocrine survival, and possibly differentiation signals yet unknown, are also at play in these cultures. 

Notably, some pathways known to play a role in Schwann cell differentiation are usually not intentionally manipulated in the current protocols. Amongst these are Notch and YAP/TAZ signaling. Notch activation can upregulate ErbB2 and, therefore, increase responsiveness to Nrg1 in Schwann cell precursors; thus, promoting their survival and further differentiation [85]. Recently, it has been shown that activation of the M2 muscarinic receptors expressed in Schwann cells advances maturation, inhibits proliferation, and upregulates promyelin genes, and that this mechanism is mediated via a downregulation of Notch signaling that, in turn, decreases ErbB2 levels [187]. As Notch signaling is involved in Schwann cell differentiation, but different actions are needed for different stages, use of molecular inhibitors of the Notch pathway, which are included in some hPSC neuralization protocols, should be considered cautiously, even more so since Notch inhibition can also block neural-crest specifier genes in hESC differentiation [36]. Next, the Hippo/YAP/TAZ signaling pathway can integrate mechanical stimuli in cells. In Schwann cells, Hippo/YAP modulates proliferation, differentiation, radial sorting, and myelination [188,189,190,191,192]. In cultured Schwann cells, the nuclear location—and thus gene-regulatory activity—of YAP/TAZ varied with cell density, substrate stiffness, and mechanical stretch [191,193,194]. As a role for mechanical forces and rigidity of the extracellular substratum is emerging in the differentiation of Schwann cells and neural crest [42,44], environmental cues in the culture system, such as cell densities and plating substrates, should be carefully considered and precisely reported. The same is true for molecular extracellular matrix signals, as collagens and laminins are known to be involved in several aspects of Schwann cell development [2,195,196]. For example, laminin 211 is able to modulate myelination via Nrg1 signaling [197], and can act synergistically with mechanical stimulation [191]. 

Furthermore, ROCK inhibitors such as Y27632 are often used in hiPSC/hESC differentiation protocols, to increase survival of cells [198], and can promote EMT transition in hiPSC [199]. Therefore, this can be useful, especially during the initial phase of differentiation. Moreover, at later stages, short-term ROCK inhibitor treatments do not seem to have adverse effects on Schwann cell differentiation and proliferation, although disturbance of myelination has been found with longer treatments in cocultures of primary Schwann cells and neurons [200].

### 3.2. Cell Sources

The targeted derivation of mammalian Schwann cells by incubation with specific compounds in vitro was first demonstrated for primary rodent neural crest cells. In 1992, Stemple and Anderson introduced the purification of neural crest cells from explant cultures by sorting for p75NTR, and they subsequently demonstrated the Schwann cell differentiation capacity of the isolated cells via incubation with forskolin in serum-containing medium [153]. Further molecular factors influencing glial differentiation were identified in isolated embryonal neural crest cells, laying the foundation for directed differentiation protocols. Since then, in vitro derivation of Schwann cells has been demonstrated for various human and non-human cell sources. This review focuses on protocols to differentiate Schwann cells from hESC [111,112,114,121] and hiPSC [117,119,121,134]. These protocols are further described in Section 3.3.1. A large portion of studies utilized MSC populations as a cell source, derived from various human [173,174,201,202,203,204,205,206] and rodent [162,163,207,208,209,210,211,212] tissues. An overview of those protocols is provided in Section 3.3.2. In rodents, MSC are mostly derived from bone marrow or adipose-derived stem cells (ADSC). The latter are easily accessible, multipotent cells from adipose tissue, which behave similarly to bone marrow-derived stem cells, regarding their characteristics and multilineage differentiation potential. Since accessibility is particularly critical in human beings, other adult tissue sources besides bone marrow and adipose tissue have also been explored, including umbilical cord, tonsils, and dermis. Furthermore, neural crest-derived cell populations can be isolated from some adult tissues [51,213] and further differentiated into Schwann cells [214,215,216,217,218]. While these cell populations exhibit neural crest cell properties, it is unclear if they represent cells directly derived from embryonal neural crest that remain in a quiescent state in adult tissues, or if they assume other adult cell fates but dedifferentiate in vitro after isolation and culture. A few other cell types have also been employed, such as skin-derived precursors [219,220,221,222], fibroblasts [177,223,224,225,226,227,228], and keratinocytes [229,230]. An overview of selected studies is provided in Section 3.3.3.

Historically, rodent MSC were the first stem cell source used for directed differentiation in vitro towards the Schwann cell lineage [163], and MSC derived from various tissues are still widely used. Compared to hiPSC and hESC, adult tissue-derived multipotent stem cells, of both of mesenchyme and neural crest origin, have several advantages: they can be drawn from easily accessible tissues, such as skin or fat, and can be cultured, expanded, and differentiated, without the need for reprogramming to the pluripotency state. In addition, MSC do not request highly specialized culture techniques and reagents, as is the case with hiPSC and hESC. Moreover, the differentiation protocols starting from hiPSC/hESC are generally rather lengthy, which is probably explained by the fact that Schwann cells specify during later stages of embryonal development and fully mature postnatally. While this entire course of development needs to be mimicked in vitro for hiPSC/hESC differentiation, adult tissue-derived multipotent stem cells already exhibit a higher degree of lineage restriction, cutting down the differentiation time towards the Schwann cell lineage. This could, possibly, also be an advantage regarding Schwann cell yield and culture purity, since MSC, in contrast to hiPSC/hESC, do not need to be guided through several multipotent precursor stages. However, MSC have a limited capacity to be expanded in vitro, which is not a problem for hiPSC/hESC lines. Therefore, the accessibility and intended application will determine the decision for a specific cell source. While hiPSC/hESC might have disadvantages regarding their differentiation time, complexity, and handling, they are highly interesting for in vitro developmental studies, as well as disease modeling applications, especially if the pathological mechanisms affecting development of the neural crest and Schwann cell lineage are to be addressed. Thinking of clinical applications such as cell transplantation, hiPSC and MSC share the advantage that they can be obtained as an autologous cell source. While hiPSC would be advantageous in this context, regarding expansion capacity, MSC might be preferred, due to their higher efficiency regarding culture and differentiation time and cost, as well as a lower risk of neoplastic behavior. Another point to consider is that Schwann cells derived from different cell sources might not exhibit the same characteristics or maturation stages, and could therefore be differentially suited to specific applications. While the same readouts for Schwann cell phenotype and maturity (i.e., marker expression, myelination capacity, neurotrophic support) are applied across studies using different cell sources, comparative studies that systematically analyze and compare the properties of cells derived from hESC, hiPSC, MSC, and other cell sources at the same time are scarce. Of note, Mukherjee-Clavin and colleagues demonstrated Schwann cell differentiation from disease and control cells of hESC, hiPSC, and fibroblast origin. They reported that Schwann cells derived from fibroblasts by direct conversion retained age-related characteristics, in contrast to cells differentiated from hiPSC and hESC, which showed upregulation of genes involved in early development. They concluded that either cell source could be more suitable for a certain disease model, depending on the pathological aspect to be investigated [119]. Further work along these lines is needed, to elucidate the common and differing properties of Schwann cells derived from various cell sources, and with this information, to identify their advantages and limitations in application. 

The following sections focus on the directed and defined approaches utilizing pluripotent stem cell lines, as well as adult tissue-derived cells with stem cell properties. We provide an overview of the protocols available to date for differentiation and characterization of Schwann cells from hESC/hiPSC, the major developments in protocols published on adult tissue-derived MSC, and some examples that use other cell sources. Primary Schwann cell culture or derivation of Schwann cells from primary embryonal direct precursor tissue will not be further discussed. Likewise, we will not consider Schwann cell differentiation of transplanted precursor cells exclusively in vivo.

### 3.3. Differentiation Protocols

#### 3.3.1. Differentiation from hiPSC and hESC

This section provides an overview of the published hiPSC/hESC-based Schwann cell differentiation protocols (summarized in Table 1), studies which were based on these but contribute crucial modifications, as well as some early publications on neural crest differentiation which demonstrated differentiation capacity to Schwann cells (summarized in Table 2). Early studies demonstrated neural crest differentiation from embryonal stem cells induced by coculture with the mouse stromal cell line PA6 [115,231] and observed the spontaneous appearance of the glial marker GFAP after long-term culture (more than 2 months) of hESC-derived neurospheres [113,232]. Such a stromal-derived inducing activity was also utilized by Lee and colleagues for neural induction of hESC by coculture with a similar mouse stromal cell line (MS5), followed by incubation of hESC-derived neural crest cells with CNTF, db-cAMP, bFGF, and Nrg1, to induce Schwann cell fate [114]. This protocol yielded cells expressing GFAP, S100b, and MBP, however only after a prolonged in vitro culture phase in the neural crest stage of at least two months. Glial differentiation potential was also reported for hESC-derived neural crest cells induced via embryoid body formation, without stromal cell coculture, by detection of some GFAP+ cells upon incubation with IGF1 and Nrg1 [233], FBS and forskolin [234], or spontaneous differentiation in long-term confluent cultures [234]. Others confirmed the Schwann cell lineage differentiation potential of hESC-derived neural crest cells and were able to also transfer this to hiPSC lines, using the same cytokine combination as Lee et al. [142], or allowing spontaneous differentiation in confluent cultures [134]. However, since the primary goal of these studies was differentiation and characterization of neural crest, and the purpose of gliogenic culture was to qualitatively demonstrate the Schwann cell differentiation capacity, these protocols are not suited for generating high-yield Schwann cell cultures. Accordingly, Wang and colleagues stated in their study that the obtained glial cells were not expandable and had low survival rates in culture [142]. 

The first dedicated Schwann cell differentiation protocol from hESC used the cell line H9 and was presented by Ziegler and colleagues. To induce neural crest as the first step of directed Schwann cell differentiation, they applied the neurosphere induction method though PA6 coculture [112]. This was followed by the work of Liu et al., who included differentiation of one hiPSC line in addition to two hESC lines (H9 as well and a second H9-derived line) [111,235]. Although their protocol did not use direct stromal cell coculture, they employed stromal-derived inducing activity for neurosphere differentiation with addition of PA6-conditioned medium. The neural crest cells generated in those protocols needed to be propagated and differentiated for prolonged periods, before becoming capable of glial differentiation, rendering both these protocols very lengthy; taking up to 100 days or more in culture, from the stem cell state to differentiated Schwann cells (Figure 3). Both protocols, Ziegler et al., 2011 and Liu et al., 2012, consisted of two distinct phases, i.e., differentiation to neural crest, followed by Schwann cell differentiation. To generate neural crest, Ziegler et al., directly seeded hESC on a confluent monolayer of PA6 stromal cells, and after 14 days of culture, picked colonies manually, which were then used to generate neurospheres. These were differentiated for another four weeks in medium containing B27 and bFGF, yielding cells of neural crest identity with glial differentiation capacity after 42 days in total. In comparison, Liu et al., started with a suspension culture of hESC/hiPSC-derived embryoid bodies, which were plated adherently after 10 days, and purified using FACS for p75NTR expressing cells four days later. The obtained cells had neural crest identity and could be expanded at this stage, before further use. For the complete process of neural crest induction and expansion, a single medium formulation was used. This comprised B27, bFGF, a ROCK inhibitor, ascorbic acid, and PA6-conditioned medium, resembling the factor composition used by Ziegler and colleagues. Before differentiating into Schwann cells, expansion at this stage had to be performed for at least 30 days, therefore yielding glia-lineage competent cells after at least 44 days in culture; comparably to the previous protocol. Subsequently, cells were incubated in adherent culture with Schwann cell differentiation factors in both protocols. In the protocol presented by Ziegler et al., the differentiation factor cocktail consisted of FBS, N2, bFGF, forskolin, and Nrg1. Cells were incubated in this medium for a total of 8 weeks, and ascorbic acid was added for the last 2 weeks. Liu et al., employed a commercially available medium formulation designed for culture of MSC that contained serum, and supplemented with only Nrg1, for a culture duration of 40 days. Both studies characterized the derived cells via PCR and immunofluorescence (IF) and confirmed the expression of a range of Schwann cell markers, such as GFAP, S100(b), PMP22, and MBP (Table 1). Ziegler et al., reported that 60% of cells were positive for the glial markers GFAP and S100 without any cell sorting steps. Liu et al., who used FACS to purify p75NTR+ neural crest cells after 14 days of differentiation, reported a yield of 85% S100b+ cells after 75 days of differentiation. Upon coculture with neurons, both studies reported a close association of stem cell-derived Schwann cells (derived from the H9 hESC line in both studies) with neurites and even some segments staining positive for myelin basic protein (MBP), characteristic of myelinating Schwann cells.

While these two protocols were a breakthrough and laid the foundation for hiPSC/hESC-derived Schwann cells, they came with the disadvantages of very long culture times and a low purity. In addition, both were dependent on non-defined stromal cell-conditioned medium. In subsequent years, considerable progress was made on neural crest differentiation from hiPSC/hESC under defined conditions [125,126,127,129,137] (reviewed by Srinivasan et al., 2019 [236]), yet it took several years until another conceptual approach and protocol was published. Indeed, the work of Kim and colleagues brought major advancements, by drastically cutting the time needed to obtain cells with Schwann cell identity, to just over a month, and reporting the first method for efficiently deriving Schwann cells without any cell sorting steps [121] (Figure 3). Instead of using stromal cell coculture or conditioned medium for neural induction of the stem cell culture, the small molecules SB431542 and CHIR99021 were used, a TGFβ/Activin/Nodal signaling inhibitor and Wnt signaling activator, respectively. In addition, Nrg1 was supplied from day six onwards. After 24 days of differentiation, a highly pure culture of Schwann cell precursors was obtained, without the need for cell sorting. Subsequently, these Schwann cell precursors were incubated in serum-containing medium with differentiation factors, Nrg1, PDGF-BB, retinoic acid, and forskolin, to promote Schwann cell maturation. Molecular differentiation factors were then withdrawn after a few days, except for serum and Nrg1, which were required for long-term culture. Cell identity was confirmed by various readouts at both Schwann cell precursor and Schwann cell stage (Table 1), and myelination capacity in vitro was also reported, albeit at low efficiency. Schwann cell identity was reached after 31–33 days of differentiation; however, the cultures for myelination studies were kept long-term, and further maturation timelines were not evaluated. The protocol was proven to work with three distinct hESC lines, as well as one hiPSC line. However, as for the previous studies of Ziegler et al., and Liu et al., the main experiments and full characterization were based on the H9 hESC line. As variations in differentiation efficiency can occur, particularly between various hiPSC lines, protocols established solely on hESC lines or confirmed in only one further hiPSC line might need further validation and, where required, modifications in order to work robustly across cell lines. A previous study, mentioned above, demonstrated that in the neural crest induction step, the performance of the same differentiation protocol can vary drastically between several hiPSC lines [142]. Indeed, work from our own lab, based on the protocol by Kim et al., also demonstrated a high variability across differentiation batches with hiPSC and showed that the efficiency and robustness of the neural crest induction step could be increased by additional activation of BMP signaling at a precisely controlled intermediate activity level [118] (Table 2). A schematic overview of this modified protocol is depicted, alongside schematic expression timelines for some typical marker proteins, in Figure 4. From day 6 to day 12, a saturating amount of BMP4 was combined with a defined concentration of BMP signaling antagonist DMH1, to achieve this intermediate activity, as described previously by Hackland et al. [140]. With this protocol, cells of neural crest identity were obtained around day 12, with Schwann cell precursors obtained around day 25. Starting on day 30 of differentiation, cells were cultured in the final medium of the protocol by Kim et al. [121], in which they can be expanded at least up to day 100. In addition, the modified protocol of Hörner et al., showed further maturation of Schwann cells in a different, serum-free medium formulation. Of the different supplementations tested, the best results were achieved in medium containing Nrg1, retinoic acid, and ascorbic acid [118]. However, as cell proliferation decreases, cultures cannot be further expanded in this medium. Therefore, this defined maturation medium formulation can be used for cells at any day of differentiation during the day 50–100 expansion phase, to achieve further maturation or if a serum-free formulation is desired for the respective experimental setup.

Another protocol by Muller and colleagues, to derive Schwann cells from hiPSC, employed a similar factor combination as Kim et al.; however, it included a few more molecules and differentiation steps [120] (Table 2, Figure 3). Others have employed a combination of CNTF, db-cAMP, and Nrg1, with or without bFGF, to generate Schwann cells from a pre-differentiated and sorting-purified neural crest culture [116,117] (Table 2, Figure 3). In addition, the differentiation potential towards Schwann cell fate was demonstrated for skin-derived precursors generated from hiPSC by incubation with forskolin and Nrg1 [237] (Table 2). Two more recent studies have also shown the usefulness of hiPSC-derived Schwann cells for disease modeling of genetic disorders affecting Schwann cells, i.e., Charcot-Marie-Tooth disease type 1A [119] and neurofibromatosis type 1 [131] (Figure 3). Mukherjee-Clavin and colleagues proved that their differentiation method worked with two healthy hiPSC lines, as well as three disease hiPSC lines and two disease hESC lines, in addition to H9 cells as a healthy hESC control [119]. For neural induction, they combined dual SMAD inhibition and the Wnt activator CHIR99021, followed by the differentiation factors db-cAMP, ascorbic acid, and serum, but they omitted Nrg1. The protocol took about 55 days to reach a matured Schwann cell stage and included a cell sorting step at the Schwann cell precursor stage (Table 1). Carrió et al., derived hiPSC lines from primary plexiform neurofibromas, i.e., benign Schwann cell tumors with mutations in the NF1 tumor suppressor gene, and differentiation into Schwann cells took about 50 days [131,238]. Differentiation towards neural crest was based on previous protocols [114,129] and employed TGFβ/Activin/Nodal inhibition combined with Wnt activation, and the subsequent differentiation towards Schwann cells was achieved with FBS, N2, forskolin, and Nrg1 [131] (Table 2). 

Regarding non-human stem cell lines (Table 4), Schwann cell differentiation was reported for cynomolgus monkey ESC [239], mouse ESC [240], and mouse iPSC [241,242]; although the latter studies using mouse iPSC did not employ an in vitro differentiation protocol, but only demonstrated differentiation towards Schwann cell lineage after transplantation in vivo, which has also been shown for human iPSC [142,243]. 

#### 3.3.2. Adult Tissue-Derived Multipotent Stem Cells 

MSC are a widely used source of multipotent cells obtainable from adult tissues. They are of mesodermal origin and can be isolated from various tissues and differentiated towards Schwann cells in vitro (reviewed by Wakao et al., 2014 [172]). Interestingly, some adult tissues such as nasal, oral, and skin tissues, also seem to harbor multipotent cells of neural crest origin [51,213]. This section reviews protocols for Schwann cell differentiation from adult tissue-derived multipotent stem cells of both mesodermal and neuroectodermal origin, since the source tissues for both of these cell populations overlap, at least partially; similar methods for Schwann cell differentiation have been employed, and the developmental origin of the isolated cells was not always characterized beyond doubt. Differentiation of adult rat bone-marrow derived MSC into Schwann cells using a defined molecular factor cocktail was first reported by Dezawa and colleagues [163] (Table 4), and several publications involving MSC-derived Schwann cells employed their protocol, or minor modifications of it. After isolation and initial culture of MSC, an induction step composed of a 24 h treatment with β-mercaptoethanol in serum-free medium, followed by three days of incubation with all-trans retinoic acid and FBS, was used to promote a switch to neural fate. This was followed by seven days of incubation with a differentiation factor cocktail, comprised of FBS, forskolin, bFGF, PDGF, and Nrg1. Slight modifications of this protocol included longer incubation times, usage of GGF-2 instead of heregulin-ß1 (which, however, are both Nrg1 isoforms), and an increased forskolin concentration [162,244,245,246,247,248,249,250,251]. Subsequently, this protocol was also demonstrated to work with rat ADSC, and was employed in several studies, mostly with an increased incubation time and forskolin concentration [162,209,211,252,253,254,255,256,257,258]. In 2007, the Dezawa lab demonstrated transfer of their protocol to human MSC [206] (Table 3), and it was used in several studies employing human MSC derived from bone marrow [259], umbilical cord [173,260,261,262], ADSC from fat tissue [204,263,264], as well as cells derived from dermis [201] and dental pulp [217]. Besides rat and human MSC, the protocol was applied to mouse MSC [208] and cynomolgus monkey MSC [265]. 

A modified version by Xu and colleagues combined an initial induction phase to neurospheres with subsequent differentiation towards a Schwann cell fate, with the same factors as used in the above-described protocols [212] (Table 4). With this method, a neural cell fate was induced (or selected for and enriched) by cultivation of floating cell clusters in serum-free medium, with the addition of EGF, bFGF, and B27 supplements. The neurosphere induction approach yielded cells of a Schwann cell phenotype from rat ADSC, simply upon withdrawal of these factors in dissociated sphere-derived cultures, without additional differentiation factors [266]. Nonetheless, most studies employing the neurosphere induction method subsequently incubated cells from dissociated spheres again with the factor combination used by Dezawa et al. (FBS, retinoic acid, forskolin, bFGF, PDGF, Nrg1). In most of these protocols, β-mercaptoethanol was not included, and retinoic acid was added directly with the other factors [174,212,267], or not at all [202,203,205,210]. While Xu et al., used rat ADSC, this method proved to also be suited for the differentiation of Schwann cells from human ADSC [205] (Table 3), or for human cells derived from umbilical cord [174], dental pulp [214], tonsils [203,268], and bone marrow [202]. The sphere induction in this protocol was used to select and enrich for a neurosphere-forming subpopulation of cells that was positive for nestin, a neural marker protein also expressed by neural crest [210]. It is likely that this step enriched a neural crest-derived cell population with neuroectodermal differentiation potential, as described above, which can also be isolated from “classic” MSC tissue such as bone marrow [267]. Therefore, it would make sense that an induction step with β-mercaptoethanol and retinoic acid is not necessary in those protocols, since this is thought to induce the fate switch in MSC before incubation with a Schwann cell differentiation factor cocktail. 

Further modifications of this factor cocktail included adding TGF-β/Activin/Nodal pathway inhibitor SB431542 [215], non-defined components such as medium conditioned by olfactory ensheathing cells [207] and coculture with neurons [202,210,269], or non-chemical factors potentially influencing differentiation such as electrical stimulation [264,270], or topographical cues [246,271,272]. Furthermore, a molecule combination of CNTF, db-cAMP, bFGF, and Nrg1, as employed by Lee and colleagues on hESC [114], was also used in combination with neurosphere induction [273].

Another source cell population for differentiation into Schwann cells are skin-derived precursors. These are less well known and characterized than MSC, but likewise exhibit stem cell properties of self-renewal and multipotency. They can be derived from adult rodent and human dermis, by culturing as floating spheres with bFGF and EGF, similarly to the neurosphere induction method described above [222,274]. These cells can be of mesenchymal or neural crest origin, depending on the exact structure they were harvested from [275,276], but either of them can be differentiated into a Schwann cell phenotype. The first dedicated protocols to generate Schwann cells from skin-derived neurospheres used incubation with the differentiation factors forskolin, Nrg1, and FBS in adherent cultures, and purification by manual picking of colonies [221,277] (Table 4). Since then, several studies have demonstrated in vitro differentiation of Schwann cells with this, or similar, methods [219,220,278,279,280,281,282].

#### 3.3.3. Other Cell Types

Besides embryonic, induced, or adult tissue-derived stem cells, in vitro Schwann cell differentiation has used human [119,177,224,225,226,227,228] and rodent [177,223] skin fibroblasts via direct conversion methods, bypassing the pluripotency stage (reviewed by Yun et al., 2022 [283]). In most studies, conversion of fibroblasts was achieved by introducing transcription factors driving neural crest and/or Schwann cell fate, mostly Sox10 [119,227] in combination with Krox20 (also known as Egr2) [177,226], and subsequent culture in Schwann cell differentiation medium containing combinations of Nrg1, forskolin, PDGF, db-cAMP, CNTF, bFGF, and FBS. Reprogramming with pluripotency transcription factors, as used in the generation of iPSC, in combination with chemical differentiation cues for Schwann cell fate was also reported [224], as well as purely chemical conversion, without the need of transgene introduction [223,225,228]. Many of these protocols included purification steps using FACS or manual colony picking. Apart from fibroblasts, primary human keratinocytes were also induced into neural crest-like cells via small molecules, and further differentiated into Schwann cells using a factor cocktail containing CNTF, bFGF, SB431542, ascorbic acid, Nrg1, and FBS for several weeks [229,230]. 

Selected publications using the cell sources described in this and the above section are listed in Table 3 and Table 4, sorted into human and non-human, respectively. We have included studies that demonstrated distinct differentiation protocols or techniques for the first time, added crucial modifications to previously published ones, or demonstrated the transfer of published protocols to another source cell type. For the latter, we have also considered the tissue origin for adult-tissue derived stem cells, as variation in differentiation potential has been shown for both MSC [284], as well as adult neural crest-derived stem cells [285], depending on the tissue they were harvested from.

## 4. Characterization

Since in vitro stem cell-derived Schwann cells usually show an immature phenotype and as differentiation efficiencies can vary, it is of high importance to thoroughly characterize the cells obtained. To provide an overview, Table 1, Table 2, Table 3, Table 4 list all methods used to characterize in vitro-derived Schwann cells in the included studies. IF, Western blotting, and analysis by PCR or transcriptomics approaches (DNA microarrays and RNA sequencing) were amongst the common methods used to characterize the marker expression of differentiated cells and intermediate stages. In addition, stereotypical cell morphology was assessed [118,204,208]. However, while morphological analysis may indicate culture state and development, human Schwann cell cultures might exhibit heterogeneous and dynamic morphologies that can significantly diverge from the typical bipolar spindle shape, due to environmental factors such as culture density, growth substrate, or medium composition [18]. At the endpoint of differentiation, the behavior and functionality of Schwann cells can be measured via in vitro and in vivo assays. In vitro assays include testing for neurotrophic factor secretion, such as NGF, BDNF, or GDNF; usually measured via ELISA [121,202,217]. Another frequently used method was the coculture with (usually primary) neurons, followed by various readouts, including induction of neurite outgrowth [120,162,174,177,202,203,212,228], alignment of Schwann cells with neurites [112,118,131,215], and myelination [111,119,121,177,202,228]. If in vitro myelination was tested, neuronal markers such as β-III-tubulin or neurofilament were stained together with myelination marker myelin basic protein (MBP), and double positive segments were quantified. The choice of appropriate markers is crucial for this readout, as there are other proteins that are associated with the myelination program but that are already expressed at earlier stages and are not definite markers of myelination, such as myelin protein zero (MPZ, P0). While in vitro coculture assays can demonstrate the capability of hPSC-derived Schwann cells to myelinate axons, this requires long-term cocultures of several weeks (up to months) and the myelination efficiencies are usually rather low. However, low in vitro myelination efficiencies were also reported for primary human Schwann cells in comparison to rodent Schwann cells or primary human oligodendrocytes [17,22]. Furthermore, some studies used in vivo transplantation assays to characterize cells, whereby in vitro differentiated Schwann cells were introduced to the crush site in rodent nerve injury models. Functional readouts included the acceleration of healing compared to control animals, as well as in vivo myelination capacity of Schwann cells [117,119,121,173,202,203,204,206]. In the following, we focus on in vitro characterization, as in vivo characterization has been reviewed elsewhere [21,51]. 

The most commonly used markers to characterize in vitro stem cell-derived Schwann cells include Sox10, p75NTR, S100(b), Gap43, GFAP, Krox20 (Egr2), PMP22, MPZ, and MBP [18,287] (Table 1, Table 2, Table 3 and Table 4). The earliest markers, Sox10 and p75NTR, already appear in neural crest. To characterize neural crest specifically, Hnk1, Sox9, Pax3, Pax7, FoxD3, Slug, Snail, or Twist have also been used in differentiation studies. Where purification of neural crest cells was included, mostly p75NTR and/or Hnk1 were used as markers, although their suitability is under discussion, as they possibly label only a subset of neural crest cells, and their expression is not restricted to neural crest [125,288,289,290]. Next to Sox10 and p75NTR, which continue to be expressed further along Schwann cell development, Pax3 persists during the transition from neural crest to Schwann cell precursors [291,292]. Then, Pax3 expression decreases as the Schwann cells mature, and in developing myelinating Schwann cells, an inverse correlation between Pax3 and MBP expression was seen [293,294]. While Kioussi et al., reported Pax3 expression in non-myelinating Schwann cells, others found that mature non-myelinating Schwann cells did not express Pax3 [291,295], except for a small stem cell-like subpopulation that persisted in the adult nerve [295]. This would make sense, as Pax3 expression positively regulates Schwann cell proliferation and seems to suppress terminal differentiation [293], and current evidence suggests that Pax3 is downregulated around the transition of Schwann cell precursors into immature Schwann cells [291]. Markers that start to be expressed during Schwann cell precursor development include Gap43, MPZ, and S100b; and MPZ/S100b further increase during the development towards immature Schwann cells. Based on current knowledge, all Schwann cell subtypes keep expressing S100b, which makes it the most widely used marker for Schwann cells. However, it is not exclusively expressed in Schwann cells, but can also be found in a variety of other cell types (e.g., other glial and neural cells, melanocytes, adipocytes, several muscle cell types) [296]. Furthermore, antibody specificity is particularly critical for S100b, as the S100 family of Ca^2+^ binding proteins includes 24 members, which are widely expressed in many cell and tissue types [297]. In general, it is crucial to combine several markers in order to unequivocally identify Schwann cells. As differentiation progresses, myelination onset markers, such as Krox20, start to be expressed during a pro-myelinating state, in which cells are not yet actively producing myelin. The latter is indicated by the expression of the mature myelination marker MBP. At the same time, expression of Gap43 and p75NTR ceases; however, both persist in the non-myelinating fate [1,2,3,26,65,78,298]. As mentioned earlier, adult non-myelinating Schwann cells are less well characterized than myelinating Schwann cells, and the marker expression likely differs, as well as between subpopulations [288]. As readouts for “mature” Schwann cells in vitro are largely based on myelination capacity and myelin-related marker expression, the characterization of potentially mature non-myelinating Schwann cells in vitro has not yet been clearly defined, as many known markers overlap with immature Schwann cell markers. A few possibly specific markers for non-myelinating Schwann cells and subpopulations were found in vivo [288,299,300,301,302], but these have not been established in vitro yet; and even in primary cultures, it is not known whether mature non-myelinating Schwann cells can be obtained with the current culturing methods [19].

Some examples of IF staining for such marker proteins are shown in Figure 5 for in vitro hiPSC-derived Schwann cells, as well as the intermediate differentiation stages, neural crest and Schwann cell precursors. These cells were differentiated according to the protocol depicted in Figure 4 [118,121], where schematic expression timelines of key markers are also shown, alongside the differentiation protocol. Figure 5A shows the marker stainings of neural crest cells after 12 days and Schwann cell precursors after 25 days of the differentiation phase, while Figure 5C show the distribution of marker proteins of differentiated Schwann cells obtained after more than 50 days of differentiation. Additionally, panel C shows the difference between cells kept in the two different media which are used in that protocol for expansion (left panel) and maturation (right panel) [118]. Besides the well-established marker proteins such as Sox10, p75NTR, Gap43, S100b, and MPZ, other, not lineage-specific markers can also be helpful for additional characterization of differentiating cultures, as shown in Figure 5A–C. These include proteins such as Sox2, which is expressed in the stem cell and initial neuralization phase, but decreases with differentiation of neural crest cells; or vimentin, which starts to be expressed with EMT. Additionally, vimentin can be helpful for assessing the morphology of cells, as well as staining of F-actin, which is illustrated by the comparative panels in Figure 5B.

## 5. Trends in Schwann Cell 3D Culture

Generally, a lot of progress has been made in the field of 3D cell culture techniques, as cultures in cell conglomerates such as spheroids or organoids have often proven to be more reliable models of cell biology and interactions than simple monolayer cultures. Similarly for Schwann cells, data suggest that 3D cultures behave differently than 2D cultures regarding disease modeling and drug testing [303]. One approach to 3D cell culture is the integration of cells into 3D matrices such as hydrogels and scaffolds, which can be designed and adapted to the needs of individual cell types. As Schwann cell cultures have been explored for a while, with the aim of clinical application in nerve regeneration, there are already substantial data and established techniques to integrate Schwann cells into 3D constructs, which can be used as nerve grafts and transplanted in vivo after induced nerve injury. Most of these studies used primary rat Schwann cells. After an initial culture phase, these were integrated into conduits manufactured from biocompatible materials, which were then used to bridge nerve injuries induced in model animals. Subsequently, nerve regeneration was measured. As these studies aimed for in vivo transplantation and often used adult Schwann cells or evaluated Schwann cell differentiation in the construct only after transplantation, we will not describe these in further detail here, but redirect to other reviews [304,305,306,307,308]. However, these studies yielded information about the culture matrices compatible with Schwann cell growth, and others have also evaluated their effect on the differentiation and behavior of Schwann cells in vitro, for which we will provide a brief overview in this section.

A huge variety of biomaterials and polymers were tested for both human and rat Schwann cells (reviewed by Gregory et al., 2021 [309]). These included hydrogels or scaffolds based on collagen [117,120,217,310,311,312,313,314,315,316], gelatin [246,317,318,319,320,321], chitosan [322,323,324], fibrin [325,326], alginate [327,328,329], basement membrane preparations such as matrigel [330,331,332,333], polyacrylamide [334,335,336], spider silk [337,338,339,340], or combinations of these [341,342,343,344,345]. Specific hydrogels were also used as bioinks for extrusion-based 3D-bioprinting, a promising technique for neural tissue engineering (reviewed by Cadena et al., 2021 [346]). Several studies demonstrated good cell viability upon bioprinting of a rat Schwann cell line, RSC96 [317,328,347,348,349,350], and primary rat Schwann cells [325,329,341,342,351]. In these studies, bioinks were often based on a mixture of gelatin and alginate, or other combinations including hyaluronic acid, chitosan, and fibrin. Furthermore, thermoplastic polymers such as polycaprolactone were used to create fiber scaffolds through electrospinning or 3D-printing [272,344,352,353,354,355,356,357,358]. Often, these materials can be further micropatterned and/or functionalized. A detailed overview of the biomaterials used and how their properties affect Schwann cell biology has been provided in recent reviews [359,360,361].

Studies that compared Schwann cell cultures from different sources in 3D versus 2D tissue culture plates found beneficial effects of 3D culture on proliferation, migration, differentiation, and neurotrophic factor production [318,326,348,352,362,363]. In Table 5, exemplary studies that showed differentiation of human and rat multipotent cells towards Schwann cells in 3D matrices are summarized. Generally, for 3D culture of primary Schwann cells, functionalization with extracellular matrix molecules such as laminin was beneficial [356,364,365,366,367,368,369], and topographical cues such as micropatterns, size of incorporated pores, or fiber diameter influenced the cell adherence, migration, alignment, elongation, and differentiation [246,271,320,355,358,370]. In materials made from electrospun fibers, uniaxially aligned fibers better promoted cell alignment, elongation, and differentiation, when compared to randomly arranged fibers [272,325,342,356,357]. Similar effects were reported from 3D-bioprinted cultures with aligned fibrin fibers in print strands [325,342]. Furthermore, polymer fibers manufactured from conductive materials and enabling electrostimulation of cells enhanced cell proliferation, migration, and differentiation [355,371,372,373,374]. Besides the composition and micropatterning, the stiffness is another important parameter of 3D culture materials. In vivo, the stiffness of peripheral nerve tissue, which changes during maturation, seems to provide instructive signals to developing Schwann cell precursors [375,376]. Some studies have selectively tested the effect of different material stiffnesses, using tunable materials, on Schwann cell differentiation and behavior, and indeed demonstrated the influence of material elasticity [191,193,246,335,366,377,378] or the application of tensile forces [318]. However, at this point, general conclusions about ideal material elasticity cannot be made, as various different elasticity ranges and cell sources were used. Nevertheless, the influence of culture matrix stiffness on Schwann cells in vitro is of great interest, as Schwann cells are known to be mechanosensitive [194,376] and integrate mechanical signals into developmental processes. Therefore, it makes sense to also consider such findings for in vitro differentiation methods, as this is already done for tissue engineering [379]. For example, mechanical stimuli and culture substrates regulate YAP/TAZ signaling in Schwann cells, which is also involved in differentiation processes [188,190,191]. In rat Schwann cells, YAP/TAZ activation through mechanotransduction acts synergistically with laminin to induce upregulation of myelin-associated factors [191,366,380]. Hence, it would be interesting to also explore such mechanisms in the differentiation of human Schwann cells in vitro.

## 6. Conclusions

Protocols to differentiate Schwann cells in vitro from pluripotent stem cell sources aim to mimic the developmental signaling events in the embryo, to derive the desired cell type. In vivo, this is achieved through different stages, i.e., (i) neural induction, (ii) neural crest specification, (iii) Schwann cell precursor specification, (iv) Schwann cell differentiation, and (v) further maturation into Schwann cell subtypes. Accordingly, in vitro differentiation protocols starting from iPSC or ESC are usually divided into two phases, i.e., the induction of neural crest identity, and a subsequent differentiation towards the Schwann cell lineage. As this targeted manipulation of cell differentiation in a dish can only be as precise as our current knowledge of the signaling mechanisms involved, the differentiation protocols develop and improve as new insights into the in vivo developmental stages emerge. However, our understanding of Schwann cell development remains incomplete and is mainly based on insights from animal models. As the mechanisms might differ between animal and human cells, it is crucial to investigate Schwann cell development more thoroughly in human cells, which has been facilitated through hiPSC. As this advancing knowledge and information on Schwann cell development is translated to cell culture and incorporated into differentiation protocols, we will hopefully achieve increasingly precise and efficient human stem cell-based methods that will work in fully defined conditions. Along these lines, this literature review is intended to serve as an overview and orientation regarding the methods employed in the protocols available to date, how they relate to development in vivo, and their respective advantages and disadvantages.

## Figures and Tables

**Figure 1 cells-11-03753-f001:**
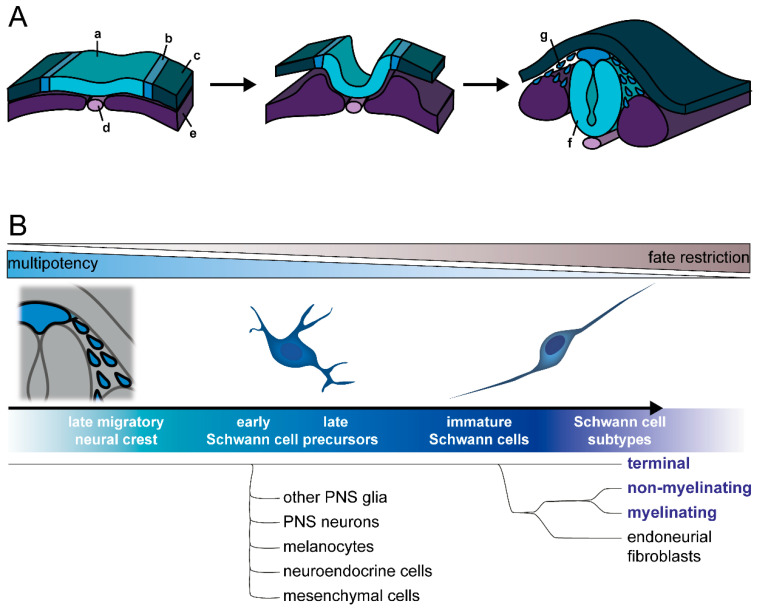
**Development of the Schwann cell lineage**. (**A**) During neurulation, the neural plate border (b) develops between the neural plate (a) and the non-neural ectoderm (c), along neighboring structures, including the notochord (d) and paraxial mesoderm (e). After the neural plate folds to build the neural tube (f), the neural plate borders merge and form the neural crest. Late neural crest cells (g) detach and migrate. (**B**) Differentiation along the Schwann cell lineage progresses through several intermediate stages, namely Schwann cell precursors and immature Schwann cells. While Schwann cell precursors are multipotent and can still differentiate into other cell types, such as melanocytes or neurons, immature Schwann cells are more fate restricted and differentiate into mature myelinating and non-myelinating Schwann cells, terminal Schwann cells of the neuromuscular junctions, as well as endoneurial fibroblasts. According to current knowledge, terminal Schwann cells segregate during differentiation prior to endoneurial fibroblasts and other Schwann cell subtypes.

**Figure 2 cells-11-03753-f002:**
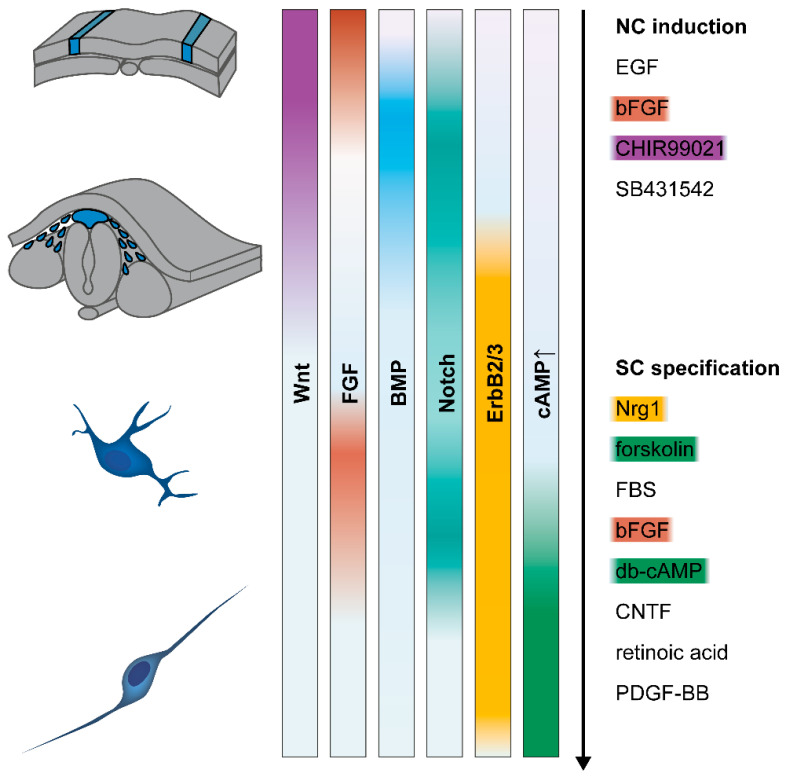
**Major signaling pathways known to be involved in Schwann cell development.** (**Left**): drawings schematically depict the major stages of Schwann cell (SC) development: neural crest (NC) formation, neural crest cell migration, Schwann cell precursor formation, and Schwann cell differentiation and maturation (from top to bottom). (**Middle**): color gradients indicate the developmental phases in which the indicated signaling pathways are known to be important. (**Right**): lists of some of the molecules frequently used for in vitro differentiation, assigned to the neural crest induction phase and Schwann cell specification phase. Color codes of the individual molecules in the right part match the corresponding activated pathways depicted in the middle.

**Figure 3 cells-11-03753-f003:**
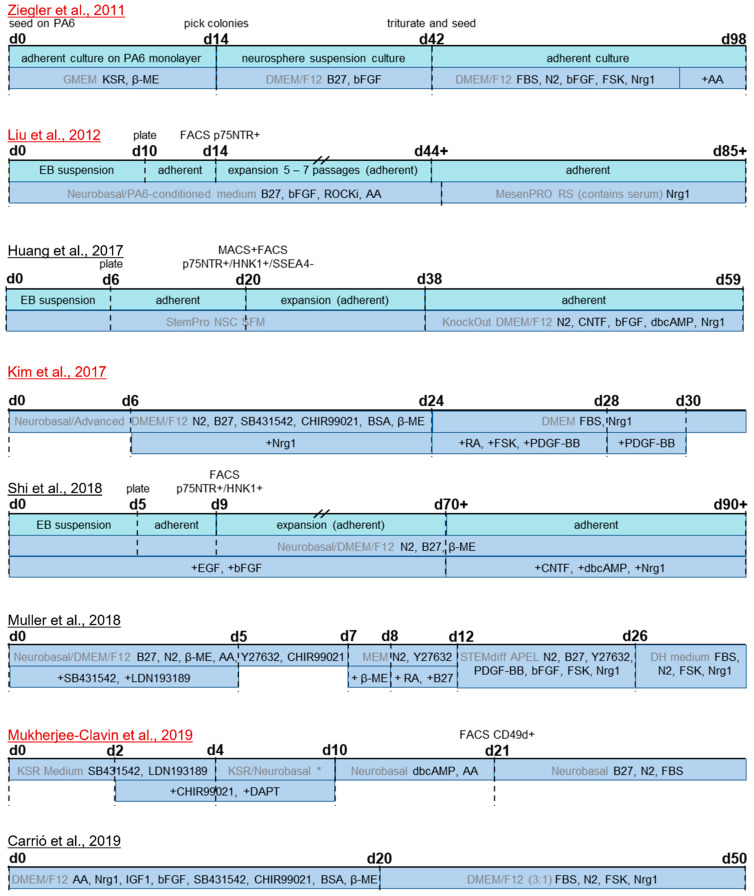
**Comparison of hiPSC/hESC based differentiation protocols**. Timelines and molecules used are shown for several principal protocols [111,112,116,117,119,120,121,131]. Publications marked in red are listed in Table 1, others are listed in Table 2. d = day of differentiation, KSR = knockout serum replacement, β-ME = β-mercaptoethanol, FBS = fetal bovine serum, FSK = forskolin, Nrg1 = neuregulin 1, AA = ascorbic acid, ROCKi = ROCK inhibitor (compound not specified), BSA = bovine serum albumin, RA = retinoic acid. * = ratio of media is successively changed: d4 KSR/NB (3:1), d6 (1:1), d8 (1:3).

**Figure 4 cells-11-03753-f004:**
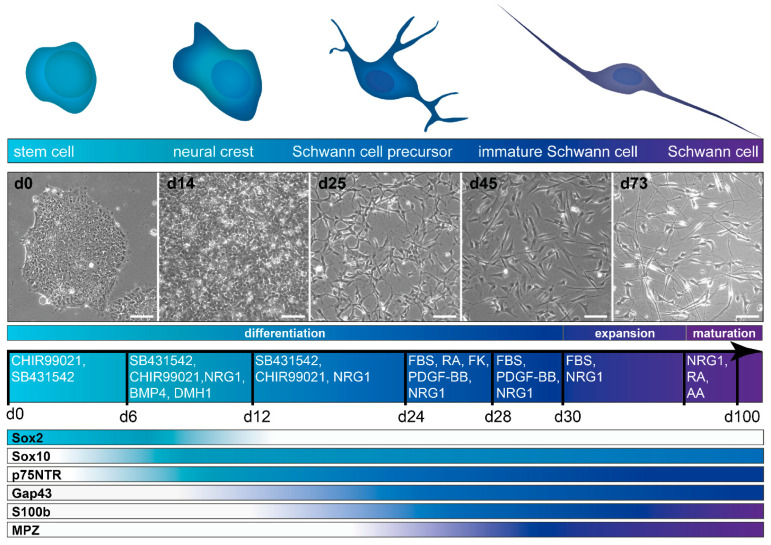
**In vitro differentiation of Schwann cells from hiPSC**. Overview of the morphology of cells at different stages during differentiation towards Schwann cells, as indicated with schematic drawings (upper panels) and representative phase contrast images of 2D hiPSC-derived cultures at the timepoints indicated (middle panels). The timeline below the microcopy images shows a schematic protocol to differentiate Schwann cells from hiPSC, as presented by Hörner et al., 2021 [118], which is based on Kim et al., 2017 [121], but used a controlled, intermediate level of BMP activity to correct for the variability of culture-intrinsic BMP signaling. RA = retinoic acid, FK = forskolin, AA = ascorbic acid. Lower panel shows schematic depiction of expression timelines of exemplary marker proteins frequently used to characterize cell stages. All microscopy images were obtained from our own work; parts of the figure were adapted with modifications from Hörner et al., 2021 [118] under CC BY 4.0 license. Scale bars, 100 µm.

**Figure 5 cells-11-03753-f005:**
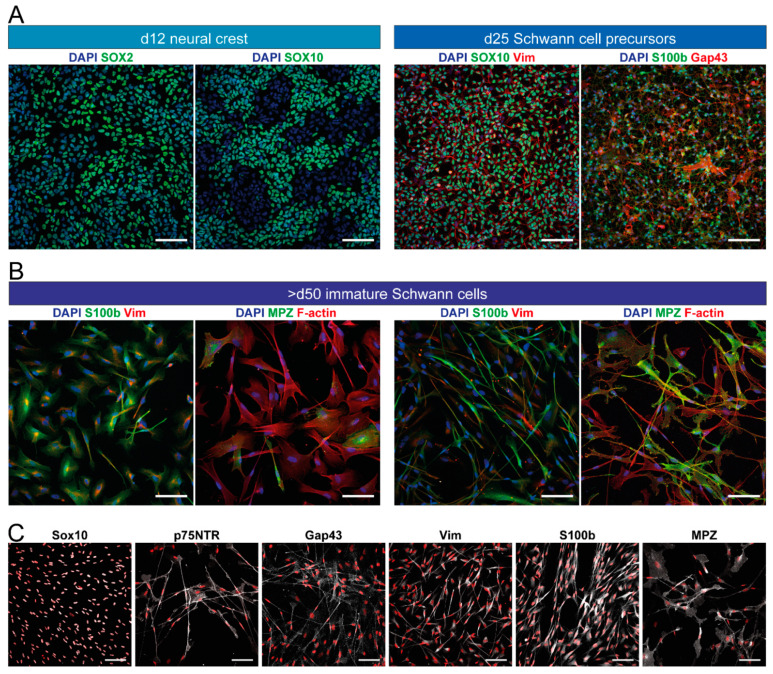
**Expression of marker proteins by hiPSC-derived Schwann cells for several stages of differentiation.** (**A**–**C**) Cells were differentiated according to the protocol presented by Hörner et al., 2021 [118], which is depicted in Figure 4. Representative immunofluorescence images of cells stained for different marker proteins (indicated) during the differentiation phase (d12; d25) and the maturation phase (>d50). All microscopy images were obtained from our own work; parts of the figure were adapted with modifications from Hörner et al., 2021 [118] under CC BY 4.0 license. (**A**) Representative marker protein signals at the neural crest stage (d12) and Schwann cell precursor stage (d25). Timepoints of fixation and color codes for immunofluorescence signals are indicated at the top of panels. Vim = vimentin. Scale bars, 100 µm. (**B**) Representative marker protein signals of cells differentiated for more than 50 days in the expansion phase (left two panels) and the maturation phase (right two panels) of the protocol. Color codes for immunofluorescence signals are indicated at the top of panels. Scale bars, 100 µm. (**C**) Immunofluorescence images of single marker proteins, as indicated at the top of panels, to show marker distribution in differentiated Schwann cells after the maturation phase. Immunofluorescence signals are shown in greyscale. Cell nuclei were labeled with DAPI and are shown in red. Scale bars, 100 µm.

**Table 1 cells-11-03753-t001:** **Studies describing primary protocols for Schwann cell differentiation from hiPSC and/or hESC.** Sorted from the most to least recent.

Study	Cell Source (Number of Cell Lines)	Duration	Characterization	Schwann Cell Markers	Sorting Steps/Yield
Mukherjee-Clavin et al., 2019 [119]	hESC (1 control, 2 disease) hiPSC (2 control, 3 disease)	approx. 55 d (21 d to SCP, up to +80 d in culture; SC maturation after 55 d in total)	IF, RT-qPCR, microarray, RNA-seq, in vitro myelination assay, in vivo transplantation assay	PMP22, SOX10, MPZ, S100b, GFAP, GalC, MBP	13.5% CD49d+ SCP at d21–23 (for hiPSC), are FACS purified for further culture; final SC yield not reported
Kim et al., 2017 [121]	hESC (3) hiPSC (1)	31 d to SC identity	IF, RT-qPCR, microarray, neurotrophic factors secretion (ELISA), in vitro myelination assay, in vivo transplantation assay	GAP43, SOX10, p75NTR, MPZ, S100b, PMP22, KROX20, GFAP	no sorting, 99% SOX10+ (SCP stage), final SC yield not reported
Liu et al., 2012 [111] Liu et al., 2014 [235]	hESC (2) hiPSC (1)	>85 d (14 d to NCC, at least 30 d expansion, 40 d to SC)	IF, RT-PCR/qPCR, microarray, in vitro myelination assay	GFAP, S100b, p75NTR, PMP22, MBP	FACS for p75NTR+ to purify NCC at d14, 85% S100b+ at d75
Ziegler et al., 2011 [112]	hESC (1)	98 d	IF, RT-PCR, neuron coculture (alignment with neurites)	GFAP, S100, p75NTR, Krox20, PMP22, MPZ, MBP	no sorting, 60% GFAP+/S100+

Abbreviations: d = days of differentiation, SCP = Schwann cell precursors, SC = Schwann cells, NCC = neural crest cells, IF = immunofluorescence.

**Table 2 cells-11-03753-t002:** **Studies demonstrating differentiation of Schwann cells from hiPSC and/or hESC, which were based on primary protocols and contributed important modifications**. Also including some neural crest protocols with first demonstrations of Schwann cell lineage differentiation capacity. Sorted from most to least recent.

Study	Cell Source (Number of Cell Lines)	Duration	Characterization	Schwann Cell Markers	Sorting Steps/Yield	Comments
Hörner et al., 2021 [118]	hiPSC (2)	31 d to SC identity (maturation after 50 d in total)	IF, morphometric analysis, neuron coculture (alignment with neurites)	SOX10, S100b, Vimentin, GAP43, MPZ	no sorting, 90% SOX10+ SCP on d19, 83.4% S100b+/80.9% MPZ+ SC	protocol based on Kim et al., 2017 [121] (Table 1); with modifications (tuned activation of BMP signaling, further maturation step)
Carrió et al., 2019 [131]	hiPSC (2 control, 2 disease)	>50 d (20 d to NCC, 30 d to SC)	IF, RT-qPCR, neuron coculture (alignment with neurites for control SC)	SOX10, p75NTR, S100b, GAP43, MPZ, PLP, PMP22, KROX20	no sorting, ~90% p75NTR+/HNK1+ NCC, final SC yield not reported	NCC protocol based on Menendez et al., 2013 [127] with modifications, disease hiPSC derived from primary neurofibroma tissue
Shi et al., 2018 [116]	hiPSC (2 control, 2 disease)	>100 d	IF, RT-qPCR, RNA-seq, in vitro myelination assay	GFAP, S100b, MPZ, MBP	FACS for p75NTR+/HNK1+ to purify NCC, final yield 74% S100b+ cells (4.8% for disease lines)	NCC protocol based on Li et al., 2015 [239] (cynomolgus monkey ESC), SC protocol based on Lee et al., 2007 [114] (hESC)
Muller et al., 2018 [120]	hiPSC (3)	26 d to SC identity (44–63 d in total until experiment endpoint)	IF, neurite outgrowth assay	S100, GFAP, p75NTR, SOX10	no sorting, yield not reported	protocol based on Kingham et al., 2007 [162]/Dezawa et al., 2001 [163] (rat ADSC/MSC, Table 4); transferred to hiPSC, with modifications for induction and maturation
Huang et al., 2017 [117]	hiPSC (not specified)	59 d	IF, neurotrophic factors secretion (ELISA), in vivo transplantation assay	SOX10, S100b, GFAP	MACS for p75NTR+, FACS for HNK1+/SEEA4- to purify NCC, final SC yield not reported	protocol based on Lee et al., 2007 [114] (hESC), with modifications for NCC and SC induction
Sugiyama-Nakagiri et al., 2016 [237]	hiPSC (1)	36 d (15 d to SKP, 21 d to SC)	IF	S100b	no sorting, yield not reported (97% SKP stage)	protocol to differentiate SKP from hiPSC, differentiation potential to SC lineage demonstrated
Kreitzer et al., 2013 [134]	hESC (1) hiPSC (4)	8 d to NCC, not given for SC	IF	GFAP	FACS for p75NTR+/HNK1+ to purify NCC, no yield reported for SC	protocol for NCC, demonstration of spontaneous differentiation into GFAP+ putative SC in mixed cultures
Wang et al., 2011 [142]	hESC (2) hiPSC (5)	>22 d to NCC, not given for SC	IF	GFAP, S100b	picking of colonies, FACS for p75NTR+; no yield reported for SC	protocol for NCC, demonstration of differentiation capacity towards SC lineage
Lee et al., 2007 [114] Lee et al., 2010 [286]	hESC (3)	28–35 d to NCC, >100 d to SC	IF	S100b, GFAP, MBP	FACS for p75NTR+/HNK1+ to purify NCC, final SC yield < 10%	protocol for NCC, demonstration of differentiation capacity towards SC

Abbreviations: d = days of differentiation, SCP = Schwann cell precursors, SC = Schwann cells, NCC = neural crest cells, IF = immunofluorescence, SKP = skin-derived precursor.

**Table 3 cells-11-03753-t003:** **Studies describing differentiation of Schwann cells from human cell sources other than hiPSC or hESC.** Sorted from most to least recent.

Study	Cell Source	Duration	Characterization	Schwann Cell Markers	Sorting Steps/Yield	Comments
Kim et al., 2020 [224]	human skin fibroblasts	35 d	IF, RT-qPCR, WB, neurotrophic factors secretion (ELISA), neurite outgrowth assay, in vitro myelination assay, in vivo transplantation assay	SOX10, GFAP, p75NTR, GAP43, S100b, PMP22, MPZ, MBP	SCP colony picking on d18, 95% SOX10+ SCP, 95% S100b+ SC	based on Kim et al., 2017 [121] (Table 1); but with direct conversion from fibroblasts instead of hiPSC
Saulite et al., 2018 [201]	human dermis MSC	8 d	IF, RT-qPCR	Sox10, p75NTR, GFAP, S100b, MBP	no sorting, 20–40% MBP+	protocol based on Dezawa et al., 2001 [163] (rat MSC, Table 4)
Bajpai et al., 2017 [230]	human epidermal keratinocytes (neonate foreskin)	7d to NCC, 35d to SC	IF, RT-qPCR	MPZ, PMP22, GFAP, S100b, Krox20	no sorting, 94% S100b+	clonal variability regarding differentiation capacity: 62.5% of clones could acquire SC fate
Mazzara et al., 2017 [177]	human skin fibroblasts, rodent skin fibroblasts (mouse embryonal/neonatal/adult, rat neonatal)	21 d	IF, RT-qPCR, RNA-Seq, in vitro myelination assay, neurite outgrowth assay	S100, O4, MPZ, GFAP, MBP	12.3% S100b+/O4+ (mouse embryonal)/5% (human) d14 → purification by FACS for O4+	full characterization only done on rodent cells, but human derived cells were also shown to induce neurite outgrowth and align with neurites
Cai et al., 2017 [202]	human MSC	56 d	IF, WB, neurotrophic factors secretion (ELISA), neurite outgrowth assay, in vitro myelination assay, in vivo transplantation assay	p75NTR, S100, MBP	no sorting, 84.9% S100+/p75NTR+	protocol based on Dezawa et al., 2001 [163] (rat MSC, Table 4)/Zhang et al., 2009 [174] (human MSC, Table 3); maturation step: coculture with rat primary neurons for SC fate commitment
Jung et al., 2016 [203]	human tonsil-derived MSC	16 d	IF, RT-qPCR, WB, neurite outgrowth assay, in vitro myelination assay, in vivo transplantation assay	p75NTR, S100b, Krox20, GFAP	no sorting, 67.6% p75NTR+	protocol based on Dezawa et al., 2001 [163] (rat MSC, Table 4)/Zhang et al., 2009 [174] (human MSC, Table 3)
Sakaue et al., 2015 [215]	human epidermal NCC from hair bulge explants	21–30 d	IF, RT-qPCR, microarray, neuron coculture (alignment with neurites)	Sox10, p75NTR, Krox20, S100b, GFAP, MPZ, MBP	no sorting, 90% Krox20+	protocol based on Dezawa et al., 2001 [163] (rat MSC, Table 4)
Martens et al., 2014 [217]	human dental pulp stem cells	18 d	IF, in vitro myelination assay, neurite outgrowth assay, neurotrophic factors secretion (ELISA)	P75, GFAP, S100	no sorting, yield not reported	protocol based on Dezawa et al., 2001 [163] (rat MSC, Table 4)
Thoma et al., 2014 [228]	human foreskin fibroblasts	39 d	IF, microarray, neurite outgrowth, in vitro myelination assay	Sox10, Krox20, PLP, GFAP, S100b, GalC, MBP	no sorting, 60% PLP+	
Tomita et al., 2013 [204]	human ADSC	18 d	IF, WB, morphometric analysis, neurotrophic factors secretion (ELISA), neurite outgrowth assay, in vivo transplantation assay	p75NTR, GFAP, S100	no sorting, yield not reported	protocol based on Dezawa et al., 2001 [163] (rat MSC, Table 4)
Razavi et al., 2012 [205]	human ADSC	16 d	IF, RT-qPCR	GFAP, S100	no sorting, 90% GFAP+/S100+	protocol based on Dezawa et al., 2001 [163] (rat MSC, Table 4)/Zhang et al., 2009 [174] (human MSC, Table 3)
Matsuse et al., 2010 [173]	human umbilical cord Wharton’s jelly-derived MSC	8 d	IF, RT-PCR, in vivo transplantation assay	Sox10, Krox20, GFAP, p75NTR, S100b, MPZ	no sorting, 98% MPZ+	protocol based on Dezawa et al., 2001 [163] (rat MSC, Table 4)
Zhang et al., 2009 [174]	human umbilical cord blood derived MSC	24 d	IF, WB, neurite outgrowth assay	GFAP, S100	no sorting, 60.8% GFAP+/S100+	protocol based on Dezawa et al., 2001 [163] (rat MSC, Table 4)/Xu et al., 2008 [212] (rat ADSC, Table 4) → neurosphere induction
Shimizu et al., 2007 [206]	human MSC	11 d	IF, in vivo transplantation assay	S100, MPZ, p75NTR, GFAP, O4	no sorting, yield not reported	protocol from Dezawa et al., 2001 [163] (rat MSC, Table 4) transferred to human MSC

Abbreviations: d = days of differentiation, SCP = Schwann cell precursors, SC = Schwann cells, NCC = neural crest cells, MSC = mesenchymal stem cells, ADSC = adipose-derived stem cells, IF = immunofluorescence, WB = Western blot.

**Table 4 cells-11-03753-t004:** **Studies describing differentiation of Schwann cells from non-human cell sources.** Sorted from most to least recent.

Study	Cell Source	Duration	Characterization	Schwann Cell Markers	Sorting Steps/Yield	Comments
Pan et al., 2021 [223]	mouse embryonic fibroblasts	>49 d	IF	S100b, GFAP	FACS purification for NCC, SC yield not reported	
Xie et al., 2017 [207]	rat ADSC	21 d	IF, RT-qPCR, WB, in vitro myelination assay, neurotrophic factors secretion (ELISA)	S100, GFAP, MBP	purification of ADSCs by FACS for CD44+, 89.5% S100+	combination of defined components and olfactory ensheathing cell conditioned medium
La Bierlein De Rosa et al., 2017 [208]	mouse MSC	24 d	IF, neurite outgrowth assay, morphometric analysis	S100, S100b, p75NTR	no sorting, 23.1% S100b+, 52% p75NTR+	based on Dezawa et al., 2001 [163]
Shea et al., 2010 [210]	rat MSC	50 d (29 d + 21 d coculture)	IF, RT-qPCR, in vitro myelination assay	S100b, p75NTR, Sox10, GFAP, MPZ, MBP	no sorting, 98.9% S100b+, 97.9% p75NTR + (after DRG neuron coculture)	based on Dezawa et al., 2001 [163]; Xu et al., 2008 [212] (neurosphere induction); addition of maturation step: coculture with rat primary neurons for SC fate commitment
Wakao et al., 2010 [265]	cynomolgus monkey MSC	9 d	IF, RT-PCR, in vivo transplantation assay	p75NTR, GFAP, MPZ, GFAP, Krox20, MBP	99% p75NTR+	protocol from Dezawa et al., 2001 [163] transferred to monkey MSC
Xu et al., 2008 [212]	rat ADSC	ca. 16 d (not precisely stated)	IF, neurite outgrowth assay, in vitro myelination assay	GFAP, S100, p75NTR	no sorting, 35% p75NTR+ (higher amount S100+/GFAP+)	combines neurospheres induction with Dezawa method
Kingham et al., 2007 [162]	rat ADSC	18 d	IF, WB, neurite outgrowth assay	GFAP, S100, p75NTR	no sorting, 42.9% GFAP+/S100+; 81.5% spindle-like morphology	protocol similar to Dezawa et al., 2001 [163]; with modifications (longer differentiation time, higher forskolin concentration)
Roth et al., 2007 [240]	mouse ESC	22–30 d	IF, RT-qPCR, neurite outgrowth assay, in vitro myelination assay	S100, GFAP, PMP22, MBP	no sorting, no yield reported	
McKenzie et al., 2006 [221] Biernaskie et al., 2006 [277]	human and rodent SKP (mouse embryo skin, rat neonate skin, human foreskin)	28–42 d	IF, in vitro myelination assay, in vivo transplantation assay	S100b, PMP22, GFAP, p75NTR, MPZ, MBP	manual SC colony picking after 2–3 weeks, >95% (rodent, after expansion of picked SC colonies; not stated for human cells)	differentiation potential demonstrated for human SKP, but purification and extensive characterization mainly done with rodent SKP
Dezawa et al., 2001 [163]	rat MSC	11 d	IF, in vivo transplantation assay	p75NTR, S100, GFAP, O4 (MBP in vivo)	no sorting, yield not reported	first report of in vitro differentiation of SC-like cells from stem cell source

Abbreviations: d = days of differentiation, SCP = Schwann cell precursors, SC = Schwann cells, NCC = neural crest cells, MSC = mesenchymal stem cells, ADSC = adipose-derived stem cells, IF = immunofluorescence, WB = Western blot.

**Table 5 cells-11-03753-t005:** **Examples for Schwann cell differentiation in 3D cultures.** Sorted from most to least recent.

Study	Cell Source	3D Culture Matrix	Comparison to 2D	Comments
Podder et al., 2022 [352]	keratinocyte- derived NC-like cells from human epidermis	electrospun polycaprolactone (PCL) aligned fibers functionalized with Nrg1	Nrg1-functionalized fibers improved differentiation	proliferation was reduced in 3D compared to 2D, expression of S100b/PLP1 increased when fibers were functionalized with Nrg1
Entezari et al., 2022 [374]	human MSC from olfactory mucosa	3D-printed PCL/ polypyrrole (PPy) conductive scaffolds	PPy-coated PCL scaffolds improved differentiation	coculture with PC12 cells, SC differentiation in 3D, SC marker expression and neurotrophic factor secretion increased on PPy/PCL compared to only PCL and 2D
Muller et al., 2018 [120]	human iPSC	collagen sponge	-	skin model, including hiPSC-derived sensory neurons, SC differentiation in 2D, matured in 3D
Bayat et al., 2016 [326]	human endometrial stem cells	fibrin gel	3D culture in fibrin gel improved differentiation	
Martens et al., 2014 [217]	human dental pulp stem cells	collagen gel	-	SC differentiation in 2D, cocultures with rat DRG neurons in 3D
Ren et al., 2013 [272]	human ESC	electrospun polyethersulfone (PES) fiber matrices	aligned fibers improved differentiation	hESC differentiation to NCC in 2D, then 2 weeks SC differentiation in 2D, then 2 more weeks either 2D or 3D; comparison of different fiber topographies
Uz et al., 2017 [246]	rat MSC	gelatin based porous conduits	best results were obtained with largest pore size and lowest stiffness	comparison between ladder-like, macroporous, and nanofibrous structures (different pore sizes and elastic moduli)
Xue et al., 2017 [356]	rat MSC	electrospun PCL fibers	aligned fibers improved differentiation	cocultures with PC12 cells and chick DRG neurons, comparison of fiber alignment and diameters, differentiation further enhanced by coating aligned fibers with laminin
Chen et al., 2015 [363]	rat MSC	fibrin matrix	increased expression of myelin-related markers and neurotrophin secretion	

Abbreviations: SC = Schwann cells, NCC = neural crest cells, MSC = mesenchymal stem cells.

## Data Availability

Not applicable.

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
