# Peer review of "Development and In Vitro Differentiation of Schwann Cells"

_cells, 2022, doi:10.3390/cells11233753_

Round 1

Reviewer 1 Report

In this manuscript, the authors reviewed the differentiation of Schwann cells from stem cells in vitro culture conditions. They have discussed in detail about the differentiation mechanisms in vitro, connecting the process with in vivo developmental condition. The paper is well written, and can be published in current form. However, I think the authors may add a schematic figure of a standard Schwann cell differentiation protocol that shows time-dependent use of agonists or inhibitors in culture, along with the expression of the cell-type markers at that stage. They also should add text describing the figure. Figure 3 shows the short protocols of several papers. I think the authors should propose their own since they have a vast knowledge in this matter. This point will interest readers of the field. Cell source is also important as the authors pointed out in the manuscript. They also recommend which cells are best for Schwann cell differentiation, and compare their efficiency with other cell types. 

I have found some tying mistakes like age 23 ‘(Error! Reference source not found). 

Author Response

The previous Figure 4 has been divided into two Figures, now Figures 4 and 5, and additional panels have been added. Figure 4 now shows a detailed scheme to illustrate the Schwann cell differentiation protocol previously published by Hörner et al., as requested. This figure depicts a timeline including differentiation molecules alongside brightfield images of principal differentiation stages and additional schematic expression timelines for some important markers. We preferred this over including the protocol in Figure 3, which was reserved for major protocol types, such as that of Kim et al. (on which the protocol by Hörner et al. was based). The new Figure 5 shows some more additional IF marker images that directly correspond to the differentiation protocol phases schematically depicted in Figure 4. Furthermore, Figure legends have been added for both new Figures. In the main text, some additional short passages were included to refer to the new Figures and describe them, on p. 15 and 26.

“(Error! Reference source not found)” mistakes were corrected. It seems these occurred by copying the manuscript text into the template word file by corrupting cross references. The manuscript has been thoroughly checked and those were removed/corrected.

Reviewer 2 Report

The authors provide a comprehensive, very well written review article on the generation of Schwann cells from different cell sources. Main focus is given to human Schwann cells that represent a tool of high interest for the development of treatment strategies for neurodegenerative conditions in the perripheral nervous system. Therfore the review is timely and will be of interest and service for a considerably large readership.

There are only three minor comments from my side:

1) the source of the photomicrographs in figure 4, B is not sufficiently clear, are these comming from the authors own work?

2) the explanation that bFGF is the old name for FGF-2 should be moved from page 12 third paragraph to the first use of bFGF in the text on page 5.

3) there are several citing errors indictaed that should be solved during revision, e.g. page 10, last paragraph, page 23, 24, 25, 26, 27, 29, 33,

Author Response

1)        All photomicrographs in former Figure 4 (new Figures 4 and 5) come from our own work. A sentence to both Figure legends was added to clearly state this.

2)         Has been moved (now on p. 6 in the revised manuscript).

3)        We believe this refers to the “(Error! Reference source not found)” cross reference issues, all of these have been resolved (see comment above to Reviewer 1). Now corrected.

Reviewer 3 Report

This review summarized the current knowledge on developmental signaling mechanisms that determine neural crest and Schwann cell differentiation in vivo and provide an overview of studies on in vitro differentiation of Schwann cells. Furthermore, a brief insight into developments regarding the culture and differentiation of Schwann cells in 3D is given. This supports the comparison and refinement of protocols and aids the choice of suited methods for specific applications.

However, the title of this manuscript is “Differentiation of Schwann cells in vitro: standards and considerations”. There are more content in In vivo development of Schwann cells. Authors should focus more on differentiation of Schwann cells in vitro, including the differences in the method of Differentiation of Schwann cells in vitro and the functional applications of different methods.

Author Response

We suggest to change the title to “Development and in vitro differentiation of Schwann cells” to better reflect all contents and focus of the Review.